# On partial randomized response model using ranked set sampling

**Azhar Mehmood Abbasi**[1,2]*, **Muhammad Yousaf Shad**[2], **Aneel Ahmed**[2]

**1** Department of IT and Computer Science Pak-Austria Fachhochschule, Institute of Applied Sciences & Technology, Haripur, Pakistan, **2** Department of Statistics, Quaid-i-Azam University, Islamabad, Pakistan

* abbasiqau2007@yahoo.com

**Data Availability Statement:** All relevant data are within the paper and its Supporting information files.

**Funding:** The authors received no specific funding for this work

## Abstract

In this paper, we propose a partial randomized response technique to collect reliable sensitive data for estimation of population proportion in ranked set sampling (RSS) scheme using auxiliary information. The idea is to increase confidence and (or) co-operation of the respondents by providing them the option of both 'direct' and 'randomized' response for the inquired sensitive question. This option is quite logical because perception of sensitive (insensitive) inquiry can vary among respondents. The properties of the proposed method are discussed and compared with existing randomized response techniques. Cost analysis is also carried out to prove supremacy of the suggested method. Finally, an application to clinical trial on AIDS is included.

## 1 Introduction

Let $Y = \{y_1, y_2, \ldots, y_M\}$ be a finite population of which estimating proportion of individuals having sensitive attribute is the prime interest. To procure such data, a respondent may be reluctant to disclose truth or provide incorrect information due to his/her privacy concerns. Therefore, such data collected through conventional (direct) ways are likely to include response-bias. In this connection [1], proposed the randomized response method with objective to collect trustworthy data by protecting privacy of the interviewee (respondent). This method, based on a randomizing device, requires interviewee to report 'yes' or 'no' as per statement pointed out by the randomizing device without revealing it to the interviewer. This technique is quite encouraging as respondent cannot be identified i.e., which question he/she has answered. After development of this pioneer work in simple random sampling (SRS) framework, different researchers have proposed a verity of methods which protect respondent's privacy by different ways and (or) give precise estimate under SRS scheme. For example, see [2–5] and references therein.

McIntyre [6] devised the RSS scheme as oppose to the conventional SRS scheme for investigation of population characteristics. This method considers ranking of the small sets of units by eyes or any other method that involves minimal cost before acquiring final sample. To choose a sample of size $m$ under RSS scheme, the experimenter first needs to identify $m$ sets of units each of size $m$ from the target population and rank the units within each set. Now, for

**Competing interests:** The authors have declared that no competing interests exist.

$i = 1, 2, \ldots, m$, select $i$th smallest unit from the $i$th set. The same process can be repeated $r$ times, if required, to obtain the sample of size $mr$. After development of RSS scheme [7], administrated its mathematical setup. To cope with the situation when direct ranking of main variable is troublesome [8], suggested the idea of indirect ranking using a concomitant variable that is highly correlated with the variable of interest. In this method, ranking is done on an auxiliary variable to choose corresponding units of the main variable. For instance, in a forest survey study, we wish to estimate height of trees; but ranking of trees may be problematic. On the other hand, the measurement on diameter of tree, which is also highly correlated with height of the tree, is easy to determine.

To select $m(\geq 2)$ units, the RSS scheme that based on an auxiliary variable $X$ can be described as: (i) construct $m$ sets of units each of size $m$ from bivariate population $(X, Y)$. (ii) In each set, obtain the exact measurement on $X$, and arrange $Y$ with respect to $X$. (iii) Select $Y$ values corresponding to $i$th $(i = 1, 2, \ldots, m)$ ordered unit of $X$ in the $i$th set. The steps (i)-(iii) can be cycled for $r$ times, if desired, to get a final sample of size $N = mr$. For more literature and application of RSS design, the interested reader is referred to the studies [9–13].

Terpstra [14] have investigated usual (insensitive) population proportion using RSS scheme and then compared with SRS rival. Following [14, 15] suggested a new proportion estimator in RSS scheme and claimed that it works better than that of [14]. Later on [16], has pointed out some bottlenecks associated with the estimator in [15] and suggested new improved estimators in RSS schemes.

Recently [17], have introduced a new sensitive proportion estimator in RSS scheme that beats SRS-based competitor suggested by [1]. In this study, keeping in view different perceptions of sensitive inquiry, we suggest a mixed (partial randomized response) method for acquiring reliable data using RSS scheme, wherein ranking is done via a continuous auxiliary variable. We discuss its properties and compare with SRS and RSS competitors given in [1, 17] respectively. The rest parts of this study are arranged as: A review of existing sensitive proportion estimators is given in Section 2. The proposed estimator, along with its properties, is developed in Section 3. Application to AIDS data is considered in Section 4. Cost analysis is done in Section 5, and final remarks are included in Section 6.

## 2 A review of existing methods

Let $Y_{1j}, Y_{2j}, \ldots, Y_{mj}$ denotes a simple random sample with replacement sampling (SRSWR) of size $m$ in $j$th cycle, for $j = 1, 2, \ldots, r$. To collect true response, each interviewee is provided with a randomizing device, say a spinner, to choose one of the following two assertion, without revealing it to the interviewer:

1. I have the sensitive attribute A

2. I do not have the sensitive attribute A

with, respective, pre-determined selection probabilities $p \neq 0.5$ and $1 - p$. Each interviewee uses the said device and reports 'yes' or 'no' as per selected statement and his/her correct status. Then 'yes' response of $i$th $(i = 1, 2, \ldots, m)$ respondent at $j$th cycle can be written as

$$\lambda = Pr(Y_i = 1) = p\pi + (1 - p)(1 - \pi)$$

Let $m_1$ be the total number of 'yes' responses from the sample size $N = mr$. [1] established the maximum likelihood (ML) estimate of $\pi$ as given by $\hat{\pi}_w = \{\hat{\lambda} - (1 - p)\}/(2p - 1)$,

$\hat{\lambda} = m_1/N, N = mr$ and showed that $\hat{\pi}_w$ is unbiased and its variance is given by

$$\text{Var}(\hat{\pi}_w) = \frac{1}{N}\left[\sigma_y^2 + \frac{p(1-p)}{(2p-1)^2}\right]$$ (2.1)

Let $Y$ be a Bernoulli variate and $X$ is a continuous insensitive auxiliary variable having cumulative distribution function (cdf) $F_X(x)$. Let $Y$ given $X = x$ also follows Bernoulli distribution i.e., $B(1, g(x))$, where $g(x)$ is a function with range (0, 1). From [17], the marginal distribution of $Y$ is $B(1, \pi)$ with parameter $\pi = \text{E}[g(X)]$, 'E' stands for expected value. In this study we assume that $\pi = \text{E}[g(\beta_0 + \beta_1 X)]$, $\beta_0, \beta_1 \in \Re$, $g(\cdot)$ is a inverse logit or probit link function, and $X$ follows normal distribution with mean 2 and variance 1 or standard uniform distribution.

Let $\{(Y_{[i]j}, X_{(i)j}): i = 1, 2, \ldots, m; j = 1, 2, \ldots, r\}$ be a RSS of size $N$, where $Y_{[i]j}$ denotes $i$th *imperfect* ranked unit in $j$th cycle, and $X_{(i)j}$ serves $i$th *perfect* ranked unit emerged in $j$th cycle. From [14], $Y_{[i]}$ is $B(1, \pi_{[i]})$ with mean $\pi_{[i]} = \text{E}[g(\beta_0 + \beta_1 X_{(i)})]$ and variance $\sigma_{y_{[i]}}^2 = \pi_{[i]}(1 - \pi_{[i]})$. [14] introduced insensitive proportion ($\hat{\pi}_t'$) in RSS and is given by

$$\hat{\pi}_t' = \frac{1}{N}\sum_{i=1}^{m}\sum_{j=1}^{r}Y_{[i]j}$$ (2.2)

Let the respondents under RSS scheme are directed to select one of the two aforesaid statements (a) and (b) by using randomizing device suggested in [1]. The respondent reports 'yes' ('no') according to the outcomes of the device and his/her actual status. Let $Y_{[i]j} = 1$ if $i$th ranked unit reports 'yes'. Then

$$Pr(Y_{[i]j} = 1) = p\pi_{[i]} + (1-p)(1 - \pi_{[i]})$$

Let $Y_{[i]1}^*, Y_{[i]2}^*, \ldots, Y_{[i]r}^*$ are independent and identically distributed (i.i.d) Bernoulli randomized responses with parameter $p\pi_{[i]} + (1-p)(1 - \pi_{[i]})$, then ML estimate of $\pi_{[i]}$ for the given data $Y_{[i]j}^*, j = 1, 2, \ldots r$ is $\hat{\pi}_{[i]} = \{\hat{\lambda}_{[i]} - (1-p)\}/(2p-1), \quad i = 1, 2, \ldots, m$, where $\hat{\lambda}_{[i]} = z_i/r$; $z_i = \sum_{j=1}^{r}Y_{[i]j}^*$ is the total number of successes observed under $i$th ranked unit. Now, overall measure of $\pi$, using the relation $\pi = \frac{1}{m}\sum_{i=1}^{m}\pi_{[i]}$, see the reference [17], is given by

$$\hat{\pi}_a = \frac{1}{N}\left[\frac{\sum_{i=1}^{m}\sum_{j=1}^{r}Y_{[i]j}^* - N(1-p)}{2p-1}\right]$$ (2.3)

and its associated variance is

$$\text{Var}(\hat{\pi}_a) = \frac{1}{N}\left[\frac{1}{m}\sum_{i=1}^{m}\sigma_{y[i]}^2 + \frac{p(1-p)}{(2p-1)^2}\right]$$ (2.4)

## 3 The proposed method

In this section, we propose a partial randomized response model to gather reliable data for estimation of population sensitive proportion under RSS scheme. This technique is applicable in the situation in which some respondents prefer 'direct response' to 'randomized response' for a sensitive inquiry. Suppose, taking into account cost and time, the experimenter decides to collect $k \geq 2$ 'direct responses' by the technique introduced by [14] and the remaining $m - k \geq 2$ units through randomized response technique suggested by [17]. Assuming that $k^2 + (m - k)^2$ identifiable units are available and following the lines of Eqs (2.2) and (2.3), we propose a

mixed (partial randomized response) model as given by

$$\hat{\pi}_k = \frac{1}{N}\left[\sum_{i=1}^{k}\sum_{j=1}^{r}Y_{[i]j} + \frac{\sum_{g=k+1}^{m}\sum_{j=1}^{r}Y_{[g]j}^* - N'(1-p)}{2p-1}\right], \tag{3.1}$$

$N' = (N - rk)$. It is obvious that $Y_{[i]j}$ and $Y_{[g]j}^*$ are Bernoulli variates with, respective, parameter $\pi_{[i]}$ and $\lambda_{[g]} = (2p-1)\pi_{[g]} + (1-p)$. It is noteworthy that $\hat{\pi}_a$, $\hat{\pi}_t'$ and $\hat{\pi}_w$ are special cases of $\hat{\pi}_k$. For $k = 0$, $\hat{\pi}_k$ reduces to $\hat{\pi}_a$. Similarly, for $k = m$, $\hat{\pi}_k$ becomes $\hat{\pi}_t'$; when $k = 0$ and $m = 1$, $\hat{\pi}_k$ simplifies to $\hat{\pi}_w$.

**Lemma**: The proposed estimator $\hat{\pi}_k$ possesses the following properties:

(i). It is an unbiased estimator i.e., $E(\hat{\pi}_k) = \pi$

(ii). It is more precise than $\hat{\pi}_a$ and $\hat{\pi}_w$ i.e., $\text{Var}(\hat{\pi}_k) \le \text{Var}(\hat{\pi}_a) \le \text{Var}(\hat{\pi}_w)$

**Proof**:
(i) From Eq (3.1), we have

$$
\begin{aligned}
E(\hat{\pi}_k) &= \frac{1}{N}\left[\sum_{i=1}^{k}\sum_{j=1}^{r}E(Y_{[i]j}) + \frac{\sum_{g=k+1}^{m}\sum_{j=1}^{r}E(Y_{[g]j}^*) - N'(1-p)}{2p-1}\right] \\
&= \frac{1}{N}\left[r\sum_{i=1}^{k}\pi_{[i]} + \frac{r\sum_{g=k+1}^{m}((2p-1)\pi_{[g]} + (1-p)) - N'(1-p)}{2p-1}\right] \\
&= \frac{1}{m}\left[\sum_{i=1}^{k}\pi_{[i]} + \sum_{g=k+1}^{m}\pi_{[g]}\right] \\
&= \frac{1}{m}\sum_{i=1}^{m}\pi_{[i]} \\
&= \pi
\end{aligned}
$$

This completes proof (i).

To prove (ii), again using Eq (3.1), we have

$$
\begin{aligned}
\text{Var}(\hat{\pi}_k) &= \frac{1}{N^2}\left[\sum_{i=1}^{k}\sum_{j=1}^{r}\text{Var}(Y_{[i]j}) + \frac{\sum_{g=k+1}^{m}\sum_{j=1}^{r}\text{Var}(Y_{[g]j}^*)}{(2p-1)^2}\right] \\
&= \frac{1}{N^2}\left[r\sum_{i=1}^{k}\sigma_{y[i]}^2 + \frac{r\sum_{g=k+1}^{m}\lambda_{[g]}(1-\lambda_{[g]})}{(2p-1)^2}\right] \\
&= \frac{1}{mN}\left[\sum_{i=1}^{k}\sigma_{y[i]}^2 + \sum_{g=k+1}^{m}\sigma_{y[g]}^2 + (m-k)\frac{p(1-p)}{(2p-1)^2}\right] \tag{3.2} \\
&= \frac{1}{N}\left[\frac{1}{m}\sum_{i=1}^{m}\sigma_{y[i]}^2 + (1-f)\frac{p(1-p)}{(2p-1)^2}\right] \\
&= \text{Var}(\hat{\pi}_a) - \frac{f}{N}\frac{p(1-p)}{(2p-1)^2}, \quad f = k/m
\end{aligned}
$$

Since the term $\frac{f}{N}\frac{p(1-p)}{(2p-1)^2}$ is always greater than or equal to zero, it is easy to observe that $\text{Var}(\hat{\pi}_k) \le \text{Var}(\hat{\pi}_a)$. Moreover [17], have showed that $\text{Var}(\hat{\pi}_a) \le \text{Var}(\hat{\pi}_w)$. This completes the proof (ii).

The relative precision (RP) of $\hat{\pi}_k$ with respect to $\hat{\pi}_j$ can be evaluated by the formula

$$\mathrm{RP}(\hat{\pi}_j, \hat{\pi}_k) = \frac{\mathrm{Var}(\hat{\pi}_j)}{\mathrm{Var}(\hat{\pi}_k)}, j = a, w \tag{3.3}$$

From [17], the limiting distribution of the proposed estimator, when $m$ is fixed and $r \to \infty$, is given by

$$\sqrt{N}(\hat{\pi}_k - \pi) \to \mathrm{Normal}\left(0, \frac{1}{m}\sum_{i=1}^{m}\sigma_{y[i]}^2 + (1-f)\frac{p(1-p)}{(2p-1)^2}\right). \tag{3.4}$$

Some interesting results can be obtained from Eq (3.4). For $k = 0$, it gives the result that presented in [17]. Furthermore, for $k = m$ or $p = 0(1)$, it becomes [14] under direct inquiry about the attribute of interest. Moreover, in the situation when both $p$ and $m$ are equal to 1, we get conventional method of direct interaction with the respondent under SRS. Note that these are not frequent occurrences; but can happen. Finally, the variance given in Eq (3.4) can be estimated by substituting $\pi_{[i]}$ by $\hat{\pi}_{[i]} = \sum_{j=1}^{r} Y_{[i]j}/r$. In this way, the estimated variance of $i$th ranked unit becomes independent of $g(\cdot)$. This validates asymptotic inference from the suggested method.

## 3.1 Comparison of proportion estimators

In this section, we examine behavior of RP given in Eq (3.3) under inverse logit link function $g$ $(\beta_0 + \beta_1 X)$ for different choices of $(\beta_0, \beta_1) \in \{(-10, 7), (-5, 5), (-2, 6)\}$, $p \in [0.1, 0.9]$ and assuming $X$ follows (i) normal distribution with parameters mean = 2 and variance = 1 i.e, N(2,1) (ii) uniform over the range 0 and 1 i.e., U(0,1). It is pertinent to mention that the RP formula is independent of $r$, hence we take different $m(= 4, 5)$ instead of $r$ to examine the performance of $\hat{\pi}_k$. Furthermore, the magnitude of correlation coefficient between $X$ and $Y$ is also computed under inverse logit link function $g(\beta_0 + \beta_1 X)$ against each above assumed pair $(\beta_0, \beta_1)$, when $X$ follows N(2,1) or U(0,1). The resultant $\rho$ values are close to {0.7, 0.6, 0.5} and {0.6, 0.4, 0.1}. Note that the results are obtained by numerical integration technique, using Mathematica Software.

In S1 Fig, we have presented $\mathrm{RP}(\hat{\pi}_w, \hat{\pi}_k)$ results obtained when inverse logit link function is used, and $X$ follows above mentioned distributions and $m = 4$. Similarly, S2 Fig depicts RP results under same setup, except, when $m = 5$. On the other hand, S3 and S4 Figs, respectively, show the function $\mathrm{RP}(\hat{\pi}_a, \hat{\pi}_k)$ for $m = 4$ and $m = 5$.

It can be observed, from S1 and S2 Figs, that RP is an increasing function of $m(k)$ and/(or) $\rho$. It is also symmetric about $p = 0.5$. In other words, for given $m$ and $\rho$, one can either choose $p$ or $1 - p$ for sensitive statement (a) without compromising RP value. For a fixed $m$, as expected, a maximum(minimum) gain in term of RP is achieved at the largest (smallest) value of $k$. Recall that at minimum $k = 0$, RP curve shows behavior of [17] estimator and it falls at the bottom position (showing less precise) as compared to the other cases when $k > 0$.

It is also easy to observe, from S3 and S4 Figs, that relative performance of $\hat{\pi}_k$ and $\hat{\pi}_a$, as expected, becomes unity when $k = 0$. However, $\hat{\pi}_k$ outperforms than $\hat{\pi}_a$ when $k > 0$. During this study, we have also examined behavior of RP values under inverse probit link function which is almost same as that under inverse logit link function. For instance, see the RP values of $\hat{\pi}_k$ vs $\hat{\pi}_a$ in S5 and S6 Figs when $m = 4, 5$. All these results strongly support our proposed estimator without compromising privacy of the respondent. In addition, $\hat{\pi}_k$ is more flexible than $\hat{\pi}_a$ and $\hat{\pi}_w$.

## 4 Application

We have first collected a real age data set of 50 AIDS patients from a local government hospital, Rawalpindi, Pakistan. Then each respondent was approached and requested to answer the question-whether he/she has had a sexual relation with any sex-worker or otherwise. The respondent was given option to either respond directly using SRS or via [1] randomizing device with $p = 0.2$ and $1 - p = 0.8$ respectively. All interviewees were also assured of their identity will never be disclosed, and made it clear that this information could help in their treatments. They were convinced enough that almost half-25 patients gave consent to disclose true information directly. After converting 'yes' and 'no' responses into binary response, we have computed a true sensitive proportion $\pi = 0.473$. We have also computed mean and variance of $X$ and its correlation with $Y$ as given by $\bar{X} = 35.88$ $\sigma_x^2 = 121.66$ and $\rho_{X,Y} = 0.417$. The main purpose of this survey was to make known of the true proportion, and compare performance of the above discussed estimators.

### 4.1 Estimation of sensitive proportion

The data acquired in this section are, now, used to estimate sensitive proportion under the proposed model and other above discussed models. To draw a sample of size $N = 8$ under partial randomized response model with $m = 4$, the following process is repeated for $r = 2$ times. Assuming $k = 0$, draw $m^2$ units and divide them into $m$ sets of size $m$. The units in different sets are ranked with respect to $X$ and then corresponding $Y$ values are found. The obtained data is displayed in S1 Table. The last column of S1 Table presents randomized response data. Similarly, the S2 Table gives layout of the data when $k = 2$. Note that in S1 and S2 Tables, $Y_{[a]bc}$ denotes $a$th imperfect ranked unit at $b$th set in $c$th cycle. However, $b$th set information is omitted, from the final data given in the last columns of the S1 and S2 Tables, for simplicity. It is pertinent to mention that we also obtained data for $m = 5$ and $r = 2$ at $k = 0, 2, 3$; but are not tabulated due to page constraints.

From S1 and S2 Tables, we have computed, using Eqs (2.3) and (3.1), $\hat{\pi}_a = 0.708$, $\hat{\pi}_k = 0.375$ at $k = 2$. To estimate sensitive proportion under [1] procedure, we drew a sample of size $N = 8$ by SRS method and got $\hat{\lambda} = 0.5$ and $\hat{\pi}_w = 0.500$.

### 4.2 Performance evaluation

In this subsection, we assess relative performance of the proposed estimator with respect to existing above discussed estimators for $m = 4, 5$ and $k = 0, 2, 3$. To this end, we have used above acquired data and variance estimate of $\hat{\pi}_h$, $h = a, k, w$ are computed for $p = 0.2$. From these numerical results, we have computed the RP values and shown in S3 Table.

The results given in S3 Table are consistent with those of plotted in S1 and S2 Figs. Thus, the partial randomized response is an efficient alternative to the existing $\hat{\pi}_a$ and $\hat{\pi}_w$, and can be used confidently to obtain accurate estimate without compromising privacy of the respondent.

## 5 Cost analysis

In survey sampling, several factors such as cost, time and accuracy or precision are taken into account to choose an appropriate sampling design. Generally, cost is main focus in almost all sampling surveys. However, this important factor was neglected in the previous Sections by assuming that there is no cost associated with the ranking of sampling units.

Following [18], we develop a cost model in RSS to assess performance of the estimators. Let $c_s$ denotes cost of stratification attached with each measured unit in ranked set sampling. In common practice, this is the cost of drawing $m - 1$ units and accomplishing judgment

ordering of the $m$ units of a set. Similarly, $c_q$ serves the cost of drawing and quantifying a unit without classification or ranking.

Now, the relative efficiency (RE) is defined as the ratio of the variance of the estimator under SRS and RSS with assumption that the total cost, say $C$, is same for both sampling schemes. Again, using the reference [18], the RE of $\hat{\pi}_k$ with respect to $\hat{\pi}_j$ is given by

$$\text{RE}(\hat{\pi}_j, \hat{\pi}_k) = \frac{c_q}{c_q + c_s} \frac{\text{Var}(\hat{\pi}_j)}{\text{Var}(\hat{\pi}_k)}, j = a, w \tag{5.1}$$

It is clear from Eq (5.1) that for fixed $c_q$ when $c_s$ varies, the magnitude of RE decreases. Moreover, maximum RE value can be gained when $c_s = 0$. To graphical present behavior of Eq (5.1), we assume different combinations of $(c_q, c_s)$ measured in dollar as {(20, 2), (20, 4)}. Note that, the values of $\text{Var}(\hat{\pi}_h)$, $h = a, k, w$ are same as obtained in Subsection 4.2.

In S7–S10 Figs, we have presented RE values of $\hat{\pi}_k$ vs $\hat{\pi}_j$ for different combinations of $(c_q, c_s)$ at $m = 4$. Likewise, the RE values of $\hat{\pi}_k$ vs $\hat{\pi}_j$ for different $(c_q, c_s)$ at $m = 5$ are displayed in S11–S14 Figs.

As expected, RE rapidly decreases as $c_s$ increases and vice-versa. However, the proposed model still remains superior. Hence, it is appropriate to estimate sensitive proportion using partial randomized response model when there is negligible (minimum) cost involved for ranking of units.

## 6 Conclusion

This study has proposed a partial randomized response model for efficiently estimating sensitive proportion under RSS scheme, wherein ranking is done via a continuous auxiliary variable. Both mathematical and numerical results supported the suggested model relative to the existing ordinary models. Moreover, the limiting distribution of the new model is also discussed and then derived some interesting results from it. The graphical representation of the numerical results revealed that the RP values between $\hat{\pi}_k$ and $\hat{\pi}_w$ are symmetric about $p = 0.5$, and became larger when $m(k)$ or magnitude of correlation between $X$ and $Y$ increases. On the other hand, RP values between $\hat{\pi}_k$ and $\hat{\pi}_a$ show similar trend, except, $\hat{\pi}_a$ also work efficiently even at low correlation ($\rho < 0.5$). Finally, the cost analysis has also been done which also advocated supremacy of the new model without compromising privacy of the respondents. Moreover, the proposed model provide the options of both direct and indirect (randomized response), therefore, it is flexible. Furthermore, the cost of direct response is far less than indirect query. Hence, proposed model is highly recommended for sensitive proportion estimation–being flexible, economical and efficient.

## Supporting information

**S1 Fig. RP $\hat{\pi}_k$ vs $\hat{\pi}_w$ under different distributions and values of $\rho$ and $p$ when $m = 4$.**
(TIF)

**S2 Fig. RP $\hat{\pi}_k$ vs $\hat{\pi}_w$ under different distributions and values of $\rho$ and $p$ when $m = 5$.**
(TIF)

**S3 Fig. RP $\hat{\pi}_k$ vs $\hat{\pi}_a$ under different distributions and values of $\rho$ and $p$ when $m = 4$.**
(TIF)

**S4 Fig. RP $\hat{\pi}_k$ vs $\hat{\pi}_a$ under different distributions and values of $\rho$ and $p$ when $m = 5$.**
(TIF)

**S5 Fig. RP $\hat{\pi}_k$ vs $\hat{\pi}_a$ for different distributions, $\rho$ and $p$ using probit link function when $m = 4$.**
(TIF)

**S6 Fig. RP $\hat{\pi}_k$ vs $\hat{\pi}_a$ for different distributions, $\rho$ and $p$ using probit link function when $m = 5$.**
(TIF)

**S7 Fig. RE $\hat{\pi}_k$ vs $\hat{\pi}_w$ under different distributions, values of $\rho$ and $p$ when $m = 4$, $c_q = \$20$ and $c_s = \$2$.**
(TIF)

**S8 Fig. RE $\hat{\pi}_k$ vs $\hat{\pi}_w$ under different distributions, values of $\rho$ and $p$ when $m = 4$, $c_q = \$20$ and $c_s = \$4$.**
(TIF)

**S9 Fig. RE $\hat{\pi}_k$ vs $\hat{\pi}_a$ under different distributions, values of $\rho$ and $p$ when $m = 4$, $c_q = \$20$ and $c_s = \$2$.**
(TIF)

**S10 Fig. RE $\hat{\pi}_k$ vs $\hat{\pi}_a$ under different distributions, values of $\rho$ and $p$ when $m = 4$, $c_q = \$20$ and $c_s = \$4$.**
(TIF)

**S11 Fig. RE $\hat{\pi}_k$ vs $\hat{\pi}_w$ under different distributions, values of $\rho$ and $p$ when $m = 5$, $c_q = \$20$ and $c_s = \$2$.**
(TIF)

**S12 Fig. RE $\hat{\pi}_k$ vs $\hat{\pi}_w$ under different distributions, values of $\rho$ and $p$ when $m = 5$, $c_q = \$20$ and $c_s = \$4$.**
(TIF)

**S13 Fig. RE $\hat{\pi}_k$ vs $\hat{\pi}_a$ under different distributions, values of $\rho$ and $p$ when $m = 5$, $c_q = \$20$ and $c_s = \$2$.**
(TIF)

**S14 Fig. RE $\hat{\pi}_k$ vs $\hat{\pi}_a$ under different distributions, values of $\rho$ and $p$ when $m = 5$, $c_q = \$20$ and $c_s = \$4$.**
(TIF)

**S1 Table. A partial randomized response real data when $m = 4$ and $k = 0$.**
(PDF)

**S2 Table. A partial randomized response real data when $m = 4$ and $k = 2$.**
(PDF)

**S3 Table. RP of $\hat{\pi}_k$ w.r.t $\hat{\pi}_w$ and $\hat{\pi}_a$ from AIDS data set.**
(PDF)

## Acknowledgments

The authors are grateful to an Academic Editor and two anonymous reviewers for providing useful comments that substantially improved the previous version of this study.

## Author Contributions

**Conceptualization:** Azhar Mehmood Abbasi.

**Funding acquisition:** Aneel Ahmed.

**Methodology:** Azhar Mehmood Abbasi.

**Software:** Azhar Mehmood Abbasi.

**Writing – original draft:** Azhar Mehmood Abbasi, Muhammad Yousaf Shad.

**Writing – review & editing:** Aneel Ahmed.

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
