## [Decision Letter · Decision Letter 0]

10 Aug 2022

PONE-D-22-10865On partial randomized response model using ranked set samplingPLOS ONE

Dear Dr. Abbasi,

Thank you for submitting your manuscript to PLOS ONE. After careful consideration, we feel that it has merit but does not fully meet PLOS ONE’s publication criteria as it currently stands. Therefore, we invite you to submit a revised version of the manuscript that addresses the points raised during the review process. Your manuscript has been assessed by two expert reviewers, whose comments are appended below. The reviewers have highlighted concerns about several aspects of the methodology and theoretical basis of some decisions. Please ensure you respond to each point carefully in your response to reviewers document, and modify your manuscript accordingly.

We look forward to receiving your revised manuscript.

Kind regards,

Joseph Donlan

Senior Editor

PLOS ONE

Journal Requirements:

5. We note you have included a table to which you do not refer in the text of your manuscript. Please ensure that you refer to Table 4-5 in your text; if accepted, production will need this reference to link the reader to the Table.

Reviewers' comments:

Reviewer's Responses to Questions

**Comments to the Author**

1. Is the manuscript technically sound, and do the data support the conclusions?

Reviewer #1: Partly

Reviewer #2: Yes

2. Has the statistical analysis been performed appropriately and rigorously? 

Reviewer #1: Yes

Reviewer #2: Yes

3. Have the authors made all data underlying the findings in their manuscript fully available?

Reviewer #1: No

Reviewer #2: No

4. Is the manuscript presented in an intelligible fashion and written in standard English?

Reviewer #1: Yes

Reviewer #2: Yes

5. Review Comments to the Author

Reviewer #1: Dear Authors, I suggest some comments which I indicate below:

-page 2: Abbasi and Shad (2021), you must specify the reference year as it is in the bibliography ((2021a) or (2021b)) so that the reader can know to which article you refer.

-page 3: Abbasi and Shad, you must add the year of publication of the article.

-page 5, 3 The Proposed method: since you explain two models in the review, Warner and Abbassi and Shad model, I think it could be interesting to compare the proposed model with Abbassi and Shad through the relative precision and also add it in the simulation part (3.1 Comparison of proportion estimators), to make the study more complete.

-page 5, 3.1 Comparison of proportion estimators: in the simulation study you use different values of m, rho, k, beta0, beta1, p and various distributions. Why have you chosen these values? Has a previous article served as a reference?

-page 5, 3.1 Comparison of proportion estimators: At the end of the section you indicate that you have used the inverse logit link function, but that you have also examined the behavior of RP with the inverse probit link function. Why have you decided to use the inverse logit link function instead of the inverse probit link function? I think that in addition to indicating with a sentence that the behavior of both is similar, it would be a good idea to include a case in which it can be seen that effectively with both functions the results are similar.

-page 6, Application: for the analyzes to be reproducible you should include the data and the codes used to obtain the results.

-page 6, 4.1 Estimation of sensitive proportion and 4.2 Performance evaluation: you refer to tables 1, 2, and 3, but these tables are not included in the article, you must include them in the text or at the end as an annex.

-page 5, 4.2 Performance evaluation: please confirm if I have understood this point correctly. The real study is carried out using the direct technique and using the Warner model and is not carried out with the method proposed by you. This is the reason why you carry out the simulation study here, to verify what would have happened with your proposed model.

-page 5, 4.2 Performance evaluation: since you explain two models in the review, Warner and Abbassi and Shad model, I think it could be interesting to compare the proposed model with Abbassi and Shad through the relative precision in the simulation part.

-page 5, 4.2 Performance evaluation: in the simulation study you use different values of m, r, k and p. Why have you chosen these values? Has a previous article served as a reference?

-page 5, 4.2 Performance evaluation: you write the following sentence at the end of the section “… and can be used confidently to obtain accurate estimate without increasing sampling cost.” I think that in this section you should not refer to the sampling cost, since you are not taking it into account at any time, in any case you could talk about the sampling variance or precision.

-page 7, 5 Cost analysis: since the cost analysis has a theoretical part and a simulation part, I would explain it before section 4 Application.

-page 7, 5 Cost analysis: since you explain two models in the review, Warner and Abbassi and Shad model, I think it could be interesting to compare the proposed model with Abbassi and Shad through the relative efficiency and also add it in the simulation part to make the study more complete.

-page 7, 5 Cost analysis: in the simulation study you use different values of cq and cs. Why have you chosen these values? Has a previous article served as a reference?

-page 7, Conclusion: the conclusions are brief. They should give more emphasis to the results obtained so that the readers use the proposed method.

-page 8, References: Horvitz, D., B. Shah, and W. Simmons (1967). (1967), the year of publication is duplicated. The title of the article should be written in capital letters.

Reviewer #2: After reviewed this article, my suggestions are

1. Some references are not listed correctly, for instance page 2 "Abbasi and Shad (2021)" a or b ??.

2. Check grammatical mistake.

3. Data are not listed and missing!

4. Some formulas are not italic.

5. I suggest to add more explanation about RSS since it is a core method in this article.

6. PLOS authors have the option to publish the peer review history of their article (what does this mean?). If published, this will include your full peer review and any attached files.

Reviewer #1: No

Reviewer #2: **Yes: **Ahmad A Hanandeh

---

## [Author Response · Author response to Decision Letter 0]

23 Sep 2022

Please see attached file "Response to Reviewers

---

## [Editor Report · Decision Letter 1]

28 Oct 2022

On partial randomized response model using ranked set sampling

PONE-D-22-10865R1

Dear Dr. Abbasi,

We’re pleased to inform you that your manuscript has been judged scientifically suitable for publication and will be formally accepted for publication once it meets all outstanding technical requirements.

Kind regards,

Beatriz Cobo

Guest Editor

PLOS ONE

Additional Editor Comments (optional):

I participated as a reviewer for the initial evaluation of this manuscript and you have revised it according to my previous recommendations. You have also resolved the comments proposed by the second reviewer.
---

## [Editor Report · Acceptance letter]

17 Nov 2022

PONE-D-22-10865R1 

On partial randomized response model using ranked set sampling 

Dear Dr. Abbasi:

I'm pleased to inform you that your manuscript has been deemed suitable for publication in PLOS ONE. Congratulations! Your manuscript is now with our production department. 

Kind regards, 

on behalf of

Dr. Beatriz Cobo 

Guest Editor

PLOS ONE